



# Nutrient transport pathways in the Lower St. Lawrence Estuary: seasonal perspectives from winter observations

Cynthia E. Bluteau[1], Peter S. Galbraith[2], Daniel Bourgault[1], Vincent Villeneuve[3], and
Jean-Éric Tremblay[3]

[1]Institut des sciences de la mer, Université du Québec à Rimouski, Rimouski, Canada
[2]Institut Maurice-Lamontagne, Fisheries and Oceans Canada, Mont-Joli, Canada
[3]Département de biologie, Université Laval, Québec, Canada

**Correspondence:** Cynthia E. Bluteau (cynthia.bluteau@uqar.ca)

**Abstract.** The St. Lawrence Estuary connects the Great Lakes with the Atlantic Ocean. The accepted view, based on summer conditions, is that the Estuary's surface layer receives its nutrient supply from vertical mixing processes. This mixing is caused by the estuarine circulation and tidal-upwelling at the Head of the Laurentian Channel (HLC). During winter when ice forms, historical process-based studies have been limited in scope. Winter monitoring has been typically confined to vertical profiles

5 of salinity and temperature and near-surface water samples collected from a helicopter for nutrient analysis. In 2018, however, the Canadian Coast Guard approved a science team to sample in tandem with its icebreaking and ship escorting operations. This opportunistic sampling provided the first winter turbulence observations, which covered the largest spatial extent ever measured during any season within the St. Lawrence Estuary and Gulf. The nitrate enrichment from tidal mixing resulted in an upward nitrate flux of about 30 nmol $m^{-2}s^{-1}$, comparable to summer values obtained at the same tidal phase. Further

10 downstream, deep nutrient-rich water from the Gulf was mixed into the subsurface nutrient-poor layer at a rate more than an order of magnitude smaller than at the HLC. These fluxes were compared to the nutrient load of the upstream St. Lawrence River. Contrary to previous assumptions, fluvial nitrate inputs are the most significant source of nitrate in the Estuary. Nitrate loads from vertical mixing processes would only exceed those from fluvial sources at the end of summer when fluvial inputs reach their annual minimum.



## 1 Introduction

Oceans and coastal seas at high-latitudes are relatively under-sampled due to their isolation and inhospitable weather. Ambitious multidisciplinary campaigns have been undertaken in the Canadian Arctic aboard the icebreaker CCGS *Amundsen*, such as CASES (Fortier et al., 2008), CFL (Barber et al., 2010) and the IPY Ccircumpolar Flaw Lead and Arctic SOLAS Experiments, but few scientific campaigns have been carried out in the St. Lawrence Estuary and Gulf during the sea-ice season (Figure 1). During this season, the CCGS *Amundsen* carries out icebreaking and ship escorting operations. In 2018, the Quebec Maritime Network initiated the Odyssée winter field program with the Canadian Coast Guard. This collaboration allowed a science team to sample alongside the Coast Guard's normal icebreaker operations. These new observations were temporally-limited but provided the first winter turbulence measurements. These turbulence measurements covered the largest spatial extent of the St-Lawrence and Gulf during any season. Previous turbulence sampling were limited to a few specific sites at the head of the Laurentian Channel (HLC, Cyr et al., 2015), downstream near Rimouski in the Lower St. Lawrence Estuary (LSLE, Cyr et al., 2011; Bourgault et al., 2012; Cyr et al., 2015), and upstream of the HLC in the Upper St. Lawrence Estuary (USLE, Bourgault et al., 2008; Richards et al., 2013). The new winter measurements were partly motivated by difficulties in modeling the physical characteristics of the water column (e.g., Smith et al., 2006b), which in turn affect biogeochemical predictions given the reliance on properties such as water temperature and turbulent diffusivity (Mei et al., 2010; Sibert et al., 2011; Taucher and Oschlies, 2011).

We were unsure what to expect from our winter mixing measurements, but the low nutrient consumption provided a better representation of the nutrient transport mechanisms than possible from summer observations. The accepted view is that nutrients in the surface layer of the Lower St-Lawrence Estuary originate from an upwelling at the HLC (Ingram, 1975, 1983; Cyr et al., 2015) and/or entrainment of deep nutrient-rich water from the estuarine circulation as fluvial waters from the USLE flows to sea (e.g., Steven, 1971). The magnitude and impact of nutrient transport caused by upwelling relative to entrainment on primary production across the whole LSLE and Gulf system is debated. Another source of nutrients is the horizontal advection of fluvial waters that drain the Great Lakes between Canada and the USA – notably the urbanised catchment upstream of Quebec City (Hudon et al., 2017).

Published field observations have typically focused on the nutrient transport from vertical mixing dynamics within the LSLE, as opposed to fluvial nutrient loads from upstream. Cyr et al. (2015) quantified with direct turbulence measurements the nitrate supply at the HLC by tidal-upwelling in late summer. Their measured nitrate vertical fluxes ranged between 0.2 and 3.5 $\mu mol\,m^{-2}\,s^{-1}$ (95% bootstrapped confidence intervals). These estimates represent the world highest reported vertical fluxes since they are an order of magnitude higher than those reported for the Mauritanian upwelling region (Schafstall et al., 2010). Cyr et al. (2015) also estimated the nitrate enrichment elsewhere in the LSLE from shear-induced mixing at the base of the nitricline via the estuarine circulation. They estimated a total of 33 and 400 mol $s^{-1}$ of nitrate being transported into the LSLE's surface layer by these two vertical mixing processes in late summer. Previous moored observations have shown that internal tides are still present at the HLC during winter (Smith et al., 2006a). However, the mixing and vertical nitrate fluxes associated with internal waves and tidal-upwelling processes have not been measured during winter.



Biogeochemical box-models have been used to evaluate the importance of vertical mixing processes in supplying nutrients
into the LSLE's surface (e.g., Savenkoff et al., 2001; Jutras et al., 2020). These box-models consider the fluvial nutrients
advected into the LSLE and assume that the estuarine circulation of the LSLE can be idealized using a few homogeneous
layers in a steady-state. Savenkoff et al. (2001) created an inverse box-model using four layers that also separated the entire
LSLE and Gulf region into eight different zones. Vertical mixing processes brought 685 mol s$^{-1}$ near the surface of the LSLE
— more than 5 times than fluvial waters (Figures 6 and 10 of Savenkoff et al., 2001). Jutras et al. (2020) revisited the nutrient
loads with a three-layer model representative of the LSLE's summer conditions. Their vertical flux of dissolved nitrogen in
the surface (1050 mol s$^{-1}$) was three times larger than the contributions from fluvial sources. Both box-modelling studies
estimated vertical fluxes of nitrogen (nitrate) that were much larger than those obtained directly by Cyr et al. (2015) in the
field. Neither considered winter conditions.

Below, we quantify the nutrient transport from diverse pathways, such as fluvial advection and vertical mixing in the Estuary.
We evaluate their relative importance for creating a nutrient inventory in the upper water column during winter, which sets-up
the subsequent spring bloom. We also extend the analysis to other seasons and revisit the importance of fluvial inputs for
supplying nutrients into the St-Lawrence Estuary throughout the year.





**Figure 1.** (a) Map of the Estuary and Gulf of St. Lawrence along with the stations visited with VMP profiles indicated by red squares and CTD profiles with black ×. Station L96 coincides with a long-term monitoring station of Fisheries and Oceans Canada's Atlantic Zone Monitoring Program (AZMP), Rimouski station. (b) Enlargement of the magenta inset in (a) to illustrate the HLC region and two Saguenay stations surveyed by helicopter in March 2018.



## 2 Study site and observations

### 2.1 Site description

The St. Lawrence seaway connects the Great Lakes to the Atlantic Ocean, with the extensive Saguenay - St. Lawrence Marine Park at the Head of the Laurentian Channel (HLC). The $\sim$400 m deep Laurentian Channel rises sharply at the HLC (Figure 1b) and denotes the upstream extent of the LSLE where the shallow USLE ends. At the HLC, the strong barotropic tide (up to 5 m amplitude) interacts with the $\approx$ 80-m sill and generates intense tidal upwelling. This interaction generates internal waves at tidal and higher frequencies, breaking lee waves, Kelvin-Helmholtz instabilities, and internal hydraulic jumps (Saucier and

Chassé, 2000). The HLC was first hypothesized by Ingram (1975, 1983) to be a turbulent mixing hot spot that supplies nutrients for primary production in the LSLE. Others considered the importance of the estuarine circulation in entraining nitrate-rich waters below the mixed surface layer into the surface layer (Steven, 1971, 1974; Sinclair et al., 1976). Throughout most of the year, the LSLE's circulation is characterized by a surface layer outflowing above a milder inflow in the intermediate layer, and even weaker inflow in the deep waters below about 150-m (see Figure 2 of Sinclair et al., 1976). The surface outflow, therefore,

becomes saltier The magnitude and impact of this nutrient transport process on primary production across the whole LSLE and Gulf system is debated.

Seawards, the LSLE progressively widens until reaching the Gulf that begins at Pointe-des-Monts (Figure 1a). At the most seaward, eastern extent of the St-Lawrence system, Cabot Strait is where warmer and saliter waters from the Atlantic enter near the bottom. These waters are formed by the warm subtropical waters transported north by the Gulf stream, which mix offshore

with cold water transported south by Labrador Current (e.g., Lauzier and Trites, 1958; Gilbert et al., 2005). These deep waters move slowly upstream because of the estuarine circulation. It takes about 3 to 4 years for water at Cabot Strait to reach the HLC (Gilbert, 2004).

### 2.2 Field measurements

The paper focuses on the field observations collected during the Odyssée winter program launched in 2018. To put these

measurements into the annual cycle context, we also present observations from existing monitoring surveys (Table 1).Namely, Fisheries and Oceans Canada monitor the biogeochemical and physical conditions in the St. Lawrence Estuary and Gulf through their Atlantic Zone Monitoring Program (AZMP). Another monitoring program, the St. Lawrence Ecosystem Health Research and Observation Network, provided additional nitrate observations for fall 2017 in the USLE.

### 2.2.1 Odyssée winter campaign of 2018

We visited 15 stations during the inaugural Odyssée program (Figure 1), which spanned from February 8 to 23 2018. We named the stations using prefixes U, L and G to designate those in the USLE, LSLE, and Gulf. These prefixes were each followed by the number of kilometres downstream from Quebec City, Tadoussac, and Pointe-des-Monts. Several sampling operations were undertaken at each station. We focus, however, on the physical observations from the conductivity, temperature, and





**Table 1.** Overview of measurement campaigns. All campaigns included CTD profiling with pumped Seabird instruments (SBE9 and/or SBE19plus V2). Bottles provided in-situ nutrients concentrations, in particular nitrate. We present phosphate and silicate only during winter. The accuracy of the SBE19's temperature and conductivity, were 0.005 °C and 0.0005 S/m. For the SBE9 and SBE-3F/SBE4-C on-board the VMP, the accuracy of temperature and conductivity were 0.001°C and 0.0003 S/m. The accuracy of the pressure sensors were 0.015% of the full-scale for the SBE9, 0.1% for the SBE19 and VMP.

| Field program | Dates | Spatial coverage | Presented data |
|---|---|---|---|
| | **Summer 2017** | | |
| AZMP | 18-19 June (USLE), 12-23 Aug | USLE, LSLE, and Gulf | Bottles and salinity |
| | **Fall 2017** | | |
| AZMP | 5-22 Nov 2017, 10 Oct 2017 (L96) | LSLE and Gulf | Bottles and salinity |
| Ecosystem Health Network | 1-3 Nov 2017 | USLE | Bottles and salinity |
| | **Winter 2018** | | |
| AZMP heli-survey† | 13 Mar 2018 | Saguenay Fjord and HLC | SBE19 and surface bottles |
| Odyssée‡ | 9-22 Feb 2018 | USLE, LSLE, and Gulf | SBE19, SBE9, VMP, and bottles |

† Surface bottles collected water at a depth of ∼ 1m. ‡ The Vertical Microstructure Profiler (VMP) carries its own Seabird CT sensors (3F/4C).

depth profiler (CTD), nutrient concentrations derived from in-situ water samples. We also collected turbulence microstructure
profiles in the LSLE and the Gulf since the USLE was too shallow for turbulence sampling (Figure 1).

At each station, a SBE9 CTD profiler (Seabird Electronics) was mounted to a rosette that was operated through the ship's moon-pool. Another pumped CTD (SBE19plus, Seabird Electronics) was regularly deployed from the vessel's side shortly after the SBE9. The SBE19-CTD's purpose was mainly to measure Photosynthetically Active Radiation (PAR) in the upper 16-m of the water column. We generally collected in-situ water samples at 10 m (i.e., 2 m beneath the ship's haul), 25 m, 50
m, 100, 150 m, 250 m and bottom, water depth permitting. These sampling depths were adjusted at shallower stations within the USLE and LSLE. At the station closest to Quebec City (U37) in 18 m depth, a single water sample was collected at 14 m depth with the rosette. Salinity was calculated from the temperature and conductivity sensors, which we present in practical salinity units (psu) herein.

All water samples obtained from the CTD-Rosette were filtered through a 0.7 $\mu$m GF/F filter using acid-washed syringes
and Swinnex. The samples were analyzed immediately on board the vessel to derive nutrient concentrations. Concentrations of $NO_3^-$ +$NO_2^-$, $NO_2^-$, $PO_4^-$ and $Si(OH)_4$ were determined using a colorimetric method adapted from Hansen and Koroleff (2007) with a Bran and Luebbe Autoanalyzer III. We calculated $NO_3^-$ concentrations by differencing $NO_2^-$ from the $NO_3^-$ and +$NO_2^-$ readings. Working standards were prepared at each station and were checked against reference standard material (KANSO CRM, lot.CH, high-Atlantic) only for the marine stations with the highest salinities. The detection limits were 0.03,
0.02, 0.03 and 0.05 $\mu$mol L$^{-1}$ for $NO_3^-$ +$NO_2^-$, $NO_2^-$, $PO_4^-$ and $Si(OH)_4$, respectively. The precision of the triplicates over the observed range of concentrations was the same as or better than these detection limits. The samples yielded nitrate, nitrite, phosphate and silicate concentrations at different depths for each station visited.





At nine of the stations located between the Saguenay Fjord and Cabot Strait (Figure 1), we collected temperature, conductivity (salinity), and turbulence profiles with a VMP-500 manufactured by Rockland Scientific Ltd. We operated the Vertical
Microstructure Profiler (VMP) from the ship's front deck. The VMP was fitted with two airfoil shear probes, two fast-response thermistors (FP07, GE Thermometics), one micro-conductivity sensor and pressure and it sampled at 512 Hz. Only data from the shear probes (5% accuracy) and the pressure sensor are presented herein. The VMP was also equipped with a high-accuracy temperature (SBE-3F) and conductivity (SBE-4C) sensors from Seabird Electronics, which sampled at 64 Hz. These measurements enabled calculating the vertical salinity and density gradients using the pressure sensor on board the VMP.

Because of the Coast Guard's operations, the VMP and CTD-Rosette sampling occurred at different phases of the tides (Figure 2). With the VMP, we were unable to cover a complete semi-diurnal tidal cycle at any station. We obtained the best temporal coverage of the tidal cycle at station G163. Fourteen VMP profiles were collected during flood tide (Figure 2). We attempted to cover another tidal cycle at station G294 on 19 February, but an ice-breaking request at the Magdalen Islands halted sampling after collecting seven profiles during ebb tide. This station was revisited on 21 February 2018 during the
ebbing tides. For all other stations, the VMP collected two to three consecutive profiles (Figure 2).

### 2.2.2  Historical monitoring surveys

Fisheries and Oceans Canada run the Atlantic Zone Monitoring Program (AZMP) multiple times each year (Therriault et al., 1998; Blais et al., 2019; Galbraith et al., 2019). Their monitoring consists of surveys in March, June, August, and November, which cover the Gulf and the LSLE (Figure 1a). Here, we present nitrate and salinities measured in summer and fall preceding
our boreal winter campaign (Table 1). For completeness, we also present nitrate and salinity observations from the fall monitoring survey of the St. Lawrence Ecosystem Health Research and Observation Network. These nitrate samples were analysed using the same procedures and equipment as those collected during the Odyssee 2018 winter field campaign.

The AZMP's ship-based fall and summer surveys provided standard CTD profiles, along with nutrient concentrations at similar depths to our winter campaign, in addition to samples closer to the surface at 2.5 and 5 m depth. The summer nitrate
measurements were collected from two different AZMP cruises (Table 1). The first cruise, during June, collected profiles in the downstream reaches of the USLE and near the HLC. The second cruise was in August and focused on the LSLE and the Gulf. Similarly to the Odyssee winter cruise, the nitrate concentrations were derived using colorimetric methods. The AZMP uses CSK standards from the Sagami Research Center, Japan and participates in inter-calibration exercises through the International Council for the Exploration of the Sea (Mitchell et al., 2002). From the CSK standards, the AZMP's nitrate concentrations have
an accuracy (rms) of 3.1, 1.7 and 1.8 % at concentrations of 5, 10 and 30 $\mu$mol L$^{-1}$, respectively. The reported values represent the triplicates' average. Their coefficient of variations were on average less than 1.2%. The majority (95%) of the 136 sets of triplicates had coefficient of variations below 3%. The summer measurements are provided to contrast with our winter nutrient observations. We use the fall measurements to estimate the nitrate inventory generated between the fall monitoring survey and our winter observations within the LSLE.

We also present observations from the winter heli-survey conducted in mid-March 2018, a few weeks after our winter program in February 2018 (Table 1). Like most years, the 2018 AZMP's winter survey was conducted by helicopter. Therefore,





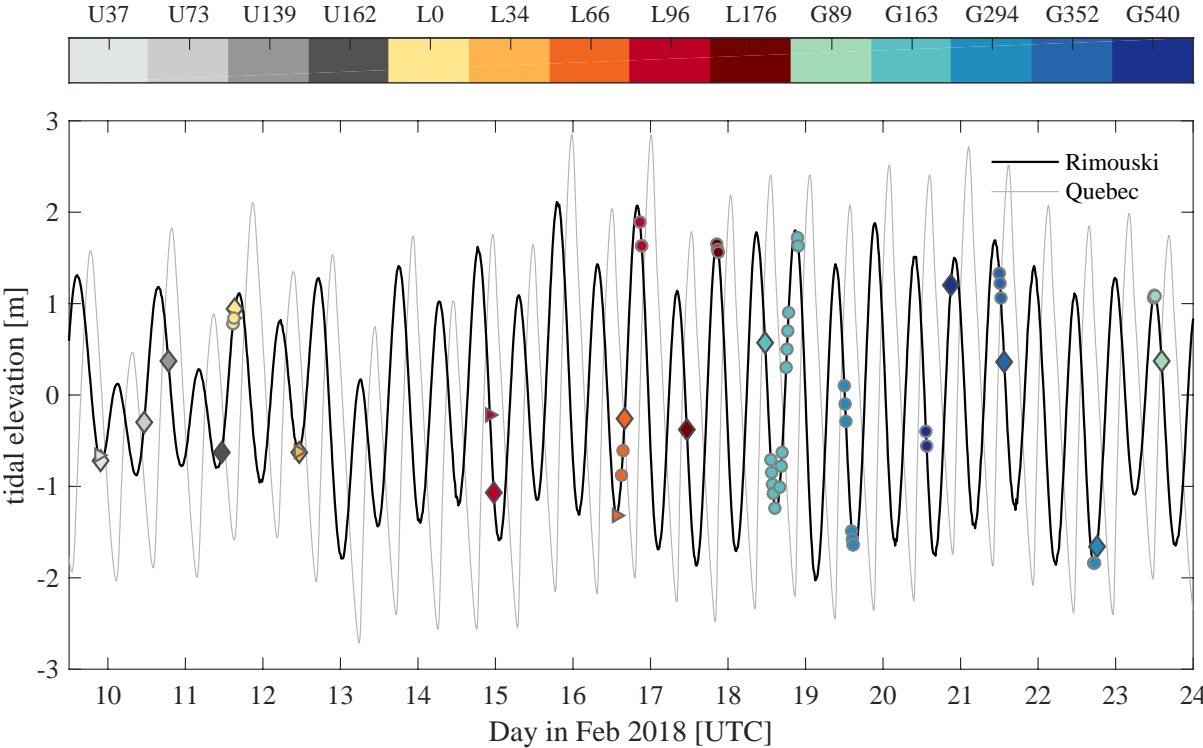

**Figure 2.** Observed tidal amplitudes at Rimouski (station #2985) and Quebec City (station #3248) maintained by Fisheries and Oceans Canada. The circles (○) and diamonds (◇) denote, respectively, when the VMP and CTD-Rosette profiled at each station designated with the colorbar. The smaller triangle (▷) denote the SBE19-CTD, which provides near-surface measurements given the rosette was deployed from the moon-pool. Tidal levels near station G163 (Grande-Vallée) precede those at Rimouski by about 15 min, while those recorded at Tadoussac (station L0) lag Rimouski by another 30 min.

sampling was limited to CTD profiles (pumped Sea-Bird Scientific SBE 19plus V2). Surface water samples were also filtered, frozen, and analyzed later for nutrient concentrations. This campaign surveyed stations within the Saguenay Fjord in addition to the St. Lawrence in mid-March (Figure 1b).

## 3 Data analysis

Our paper focuses on establishing the main transport pathways for nutrients in the LSLE. In this region, nitrate concentrations are generally lower than silicate and, therefore, more likely to limit primary production (Tremblay et al., 1997; Jutras et al., 2020). Our subsequent analysis thus focuses on tracking nitrate inputs into the LSLE's surface layer. We treat the upper-75 m of the LSLE as a box. The box receives fluvial (horizontal) inputs of nitrate from the USLE and vertical inputs entering its base through mixing processes (Figure 3a). These mixing processes include tidal-upwelling at the HLC and shear-induced mixing processes elsewhere in the LSLE. Only mixing at the base of the box is considered, extending across the upper 75-m of the





water column since mixing within the box redistribute the fluvial nitrate entering from the USLE. This shear-induced mixing at the base of the box is associated mainly with the estuarine circulation and occurs at the interface between the different layers of water. The vertical mixing caused by the estuarine circulation is more persistent than tidal-dependent upwelling mixing at the

HLC. The vertical nitrate fluxes will be quantified using the techniques described in §3.2, and applied to the respective surface areas of the HLC and the LSLE. These vertical nitrate loads will then be compared to the horizontal fluvial inputs described in § 4.3.1. Our analysis obviously ignores consumption of nutrients within the system, or any export. Our goal, however, is to compare the fluvial nitrate loads with those entering the box from vertical mixing processes.

### 3.1 Fluvial nitrate loads

We estimated the advected (horizontal) fluvial contributions of nitrate into the LSLE using nitrate concentrations and flow rates at Quebec City. The nitrate loads were thus calculated at the upstream extent of the USLE, where water is fresh. We assume that the fluvial loads in the LSLE are representative of those at Quebec City, or alternatively, that our box includes the USLE. The fluvial nitrate loads, expressed in $\mathrm{mol\,s^{-1}}$, in the St. Lawrence River is calculated from:

$$M_{\mathcal{N}} = \frac{Q\mathcal{N}}{1000} \tag{1}$$

using the flow rate $Q$ ($\mathrm{m^3\,s^{-1}}$) and the nitrate concentration $\mathcal{N}$ ($\mathrm{mmol\,m^{-3}}$) at Quebec City. These flow rates were calculated on a daily basis using the inverse modelling techniques recently developed by Bourgault and Matte (2020a, b). The approximate error on these daily and monthly flow rates are $0.25 \times 10^4\ \mathrm{m^3\,s^{-1}}$ and $0.16 \times 10^4\ \mathrm{m^3\,s^{-1}}$, respectively (Bourgault and Matte, 2020a). This method of estimating flows replaces the older and less accurate method of Bourgault and Koutitonsky (1999), which is currently issued on a monthly basis by the Government of Canada.

The historical dissolved nitrate concentrations at Quebec City were digitized from published sources (Figure 4 of Hudon et al., 2017). These measurements were collected every month and sometimes weekly from 1995 to 2011. We also estimated the fluvial nitrate loads during our winter campaign. The nitrate concentrations $\mathcal{N}$ nearest to Quebec City (station U37) were used in (1). The sampled water was almost fresh with salinities below 0.5 psu and dissolved nitrate concentrations of 26.43 mmol $\mathrm{m^{-3}}$. The application of equation 1 to the historical measurements provided long-term statistics on the fluvial nitrate

loads entering the USLE, which are presented in section 4.3.1.

### 3.2 Turbulent vertical nitrate fluxes

We combined the VMP's turbulence profiles with the nutrients measurements to obtain vertical fluxes along the St. Lawrence during winter via:

$$F_N(z) = -K(z)\frac{\partial \mathcal{N}(z)}{\partial z}, \tag{2}$$

where $z$ is the height above the free surface. The mixing rates $K$ were derived from the VMP's measurements (§3.2.1), while the vertical background concentration gradients $\partial \mathcal{N}/\partial z$ were derived for nitrate from the VMP profiles via a nitrate-salinity relationship developed by analysing the rosette's water samples (§3.2.2).

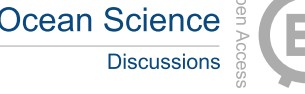



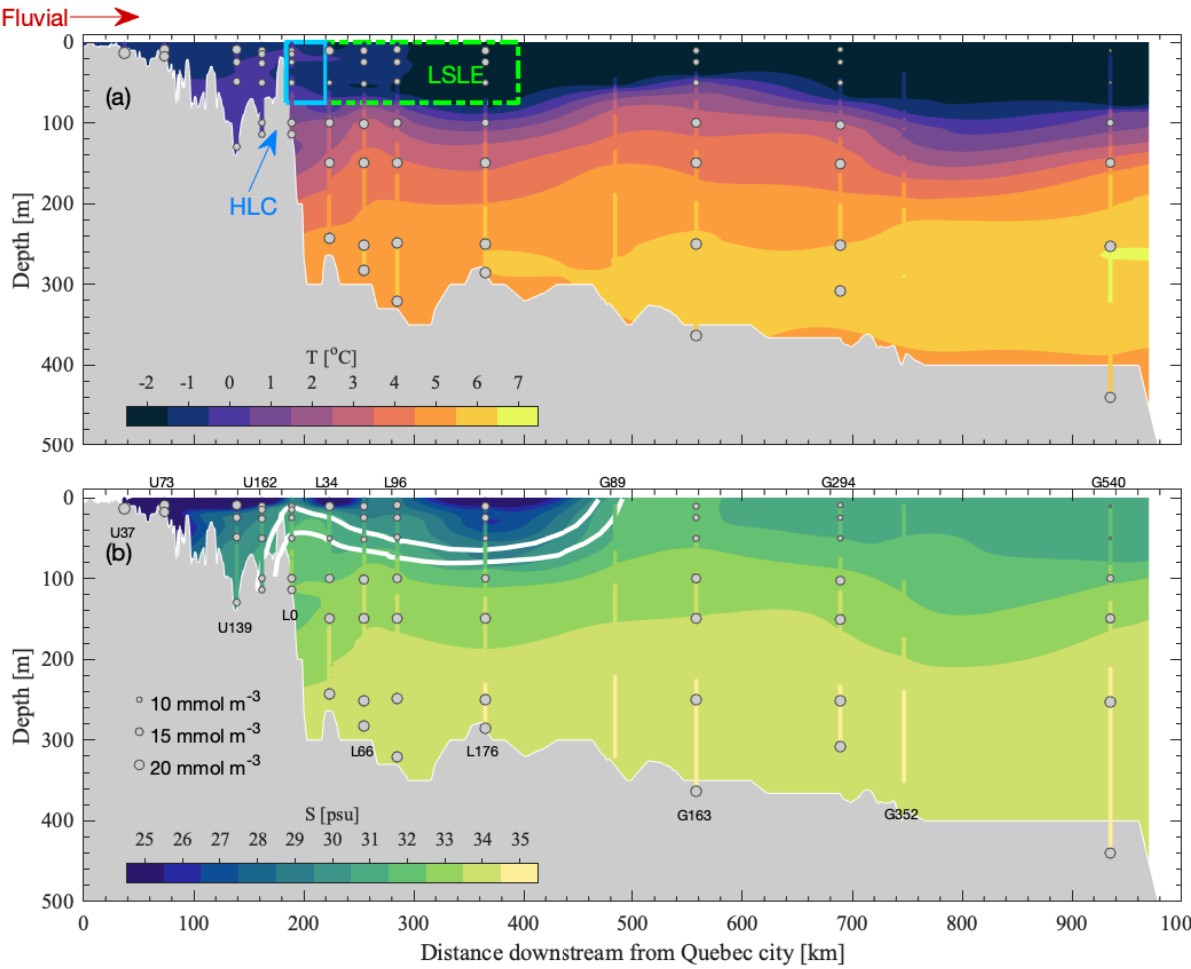

**Figure 3.** (a) Temperature and (b) salinity measured along the St. Lawrence during the field campaign. The grey circles are scaled with the nitrate concentrations of the rosette water samples in Figure 4a. The extent of the HLC and the LSLE are denoted in (a) by the cyan and green boxes, respectively. Tidal-upwelling processes are confined to the HLC. Elsewhere in the LSLE, vertical mixing at the base is caused by the estuarine mixing. The thick white lines in (b) delineate the salinity limits of 31.2 and 31.9 psu used for developing the nitrate-salinity relationships (equation 4). Water above the upper line ($S = 31.2$ psu) originated from the USLE. Water below the lower line ($S = 32$ psu) originated from the Gulf.

### 3.2.1 Diapycnal mixing rates $K$

The most commonly-used model for estimating $K$ was proposed by Osborn (1980) for shear-induced mixing:

$$K = \Gamma \frac{\epsilon}{N^2} \tag{3}$$





and requires estimating the rate of dissipation of turbulent kinetic energy $\epsilon$ and the background buoyancy frequency $N = \sqrt{(-g/\rho)(\partial\rho/\partial z)}$. A constant mixing efficiency $\Gamma = 0.2$ is often assumed, despite mounting evidence that it varies (e.g., Monismith et al., 2018). Several parametric models have been proposed (e.g., Ivey et al., 2018), and debated (e.g., Gregg et al., 2018), to relate $\Gamma$ with external parameters.

During the winter field program, the temperature gradients were generally gravitationally unstable, i.e., cold water overlaying warmer water. These unstable temperature gradients were stabilized by the salinity gradients. In these situations, double-diffusive convection (DDC) is possible. However, our observations lacked the presence of distinctive large ($\sim$ several meters high) steps that are typically suggestive of DDC. Even when DDC dominates in weakly sheared flows, equation 3 can be used to estimate $K$ by increasing $\Gamma \sim 1$ (see Hieronymus and Carpenter, 2016; Polyakov et al., 2019). In these situations, buoyancy
is the main source of mixing. Our turbulence levels were much higher than those reported by Polyakov et al. (2019) and the strong tides are more conducive to shear-induced mixing than quiescent DDC mixing. We thus assume the custom value of $\Gamma = 0.2$ for shear-induced mixing. Our chosen $\Gamma$ is consistent with field observations at low $\epsilon/(\nu N^2)$ (e.g., Holleman et al., 2016; Monismith et al., 2018). Here, $\nu$ represents the kinematic viscosity of seawater, while the ratio $\epsilon/(\nu N^2)$ is proportional to ratio of largest and smallest turbulent overturns in a stratified fluid. During our field campaign, $\epsilon/(\nu N^2)$ were 95% of the
time less than 500.

    To obtain the mixing rate $K$, we estimated $\epsilon$ using the methods described by Bluteau et al. (2016). Each profile was split into 4096 samples (8 s) that overlapped by 50% before computing the velocity gradient spectra. The VMP's profiling speed was derived from its pressure sensor to convert the spectra between the frequency and wavenumber domain. We applied the multivariate technique of Goodman et al. (2006) to remove motion-induced contamination from these spectra, which were then
integrated over the viscous subranges to obtain $\epsilon$. The VMP typically profiled at around 0.65 m s$^{-1}$, thus providing turbulence estimates at a resolution of about 2.5 m given the 50% overlap when splitting the cast. Turbulence estimates near the end and the beginning of a cast were discarded because of the VMP's deceleration and acceleration, respectively. We also discarded estimates within 25 m of the surface because of the turbulence induced by the ship. To derive the density gradients $\partial\rho/\partial z$, we relied on the high-accuracy temperature and conductivity sensors (SBE-3F and SBE-4C) aboard the VMP. We first low-pass
filtered these signals with a Butterworth filter using a cutoff period of 8 s. We then centred-differenced these smoothed profiles before averaging them over the same segments used for getting $\epsilon$ — yielding the mean vertical gradients necessary for using equations 2 and 3.

### 3.2.2   Proxy nitrate concentrations for estimating $\partial\mathcal{N}/\partial z$

Proxy nitrate concentrations are required to derive vertical nitrate gradients from the VMP's measurements (equation 2). This
proxy made use of the winter dependency between nitrate and salinity. When limited amounts of nutrients are being consumed (or generated), mixing and advection processes govern the spatial distribution of nutrients. Their concentrations vary linearly with salinity, as reflected by our winter observations in Figure 4f. A concave nitrate-salinity curve, such as observed during summer and fall 2017, indicates nitrate was being consumed along the USLE (gray points in Figure 4d,e). These trends in the nitrate-salinity diagrams are supported by incubation experiments that quantify the nitrate consumption (Villeneuve, 2020).



Nitrate consumption in the lower reaches of the USLE and in the LSLE was more than an order of magnitude higher in summer than during winter. Nitrate consumption downstream of Quebec City was less than 0.50 nmol m$^{-3}$ s$^{-1}$ and about half as much in the LSLE (Figure 16 of Villeneuve, 2020). Hence, a nitrate-salinity relationship is thus justified to estimate nitrate from the VMP profiler's salinity measurements during winter.

Nitrate concentrations in winter depended on salinity but also on the location in the St. Lawrence (Figure 4). The nitrate
variations are associated with the water masses in the region. The temperature-salinity diagram suggests two, possibly three, significant water masses across the region (Figure 4a). The first water mass is nutrient-rich waters from the USLE that mix in the LSLE before eventually mixing with nutrient-poor surface water downstream in the Gulf (Figure 4d). This mixing resulted in nitrate $\mathcal{N}$ concentrations decreasing with salinity for $S < 31.2$ psu. For higher salinities, $S > 31.9$ psu, which includes samples deeper than 50-m in the LSLE and almost all samples in the Gulf, nitrates increased proportionally with salinity (Figure
4f). This vertical nutrient distribution resembles the expected waters exposed to the open ocean, which we associate with the region's second water mass. Evidence of a third water mass is featured between 31.2 and 32 psu for nitrate, phosphate, silicate, and in particular, the temperature-salinity diagram of LSLE stations (Figure 4). The salinity of 31.2 psu corresponds to a local subsurface maximum in water temperatures at station L0, where Saguenay Fjord enters the St.-Lawrence, and downstream at L34. The heli-survey a month later confirms that the Saguenay Fjord discharged fresher water near this station (Figure 4a). The
surface water samples at the head of the Fjord had much lower phosphate ($\sim 0.2$ mmol m$^{-3}$) and nitrate ($\sim 10$ mmol m$^{-3}$), than water with comparable salinity in the USLE (Figure 4). This third water mass created a more rapid decrease in nitrate between the 31.2 to 32 psu salinity range before increasing again due to mixing with Gulf waters within the LSLE (Figure 4f).

From the observed water masses in winter, we created three separate nitrate-salinity relationships to obtain proxy nutrient concentrations the VMP's salinity measurements:

$$
\quad \mathcal{N} \, [\text{mmol m}^{-3}] = \begin{cases} -0.70S + 34.9 & \text{for } S \leq 31.2 \text{ psu,} \\ -4.37S^2 + 272.3S - 4230 & \text{for } 31.2 < S \leq 31.9 \text{ psu,} \\ 4.84S - 144.2 & \text{for } S > 31.9 \text{ psu or in the Gulf.} \end{cases} \quad (4)
$$

The first relationship, applicable for $S < 31.2$ psu, reflects the nutrient-rich water in the USLE mingling with saltier water downstream. The third relationship included all samples with $S > 31.9$ psu. It was applied to turbulence profiles within the Gulf except for the deep data near Cabot Strait (station G540, $\gtrsim$250m). This data corresponds to a fourth water mass, which we exclude from our vertical nitrate fluxes analysis. The second relationship links the other two using a quadratic fit to the samples
outside of the Gulf within the range $31.2 < S \leq 31.9$. This relation reflects contributions from the Saguenay Fjord, which was applied solely to turbulence profiles collected in the LSLE. Outside of the $31.2 < S \leq 31.9$ range, we applied the first or third relationship over their applicable salinity ranges. Typically, the first relationship was applied to surface waters in the LSLE that originated from the nutrient-rich USLE upstream. In contrast, the third relationship was used for deeper waters entering from the Gulf. A mean relative error of 5.5% was obtained for the predicted $\mathcal{N}$ after applying equation 4 to all the 64 samples in
Figure 4f. The poorest agreement is with the low surface nitrate concentrations measured in the Gulf, particularly the furthest





downstream near Cabot Strait (station G540 in Figure 4f). The proxy nitrate concentrations calculated from (4) were then used to estimate the vertical nitrate gradients necessary for obtaining vertical fluxes from equation 2.

### 3.2.3 Vertical mixing contributions of nitrate

To convert the vertical nitrate fluxes into mass loadings, we rely on the same techniques as Cyr et al. (2015) to determine the

tidal-upwelling zone. This mixing process creates a surface signature of cooler water during summer (see Figure 13 of Cyr et al., 2015), and likely coincides with the winter polynia observed during regular monitoring (see Figure 10 of Galbraith et al., 2019) and our winter cruise (Figure 4a). Their estimated surface area was approximately 100 to 200 km$^2$. For our box-analysis, we require the surface area at the 75-m isobath rather than at the air-sea interface. At this isobath, the entire LSLE covers about 6000 km$^2$, whereas the HLC is about 200 km$^2$ (Figure 3a). We thus exaggerate the impact of the HLC's tidal-upwelling on

transporting nitrate into the surface layer by using the area upper bound. For the remaining 97% of the LSLE's area, we apply the average vertical nitrate fluxes obtained outside of the HLC to the base of our box.



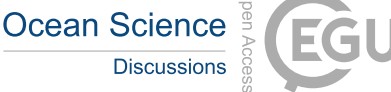

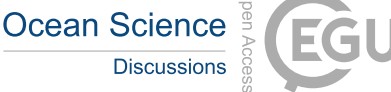

**Figure 4.** Winter 2018 observations of (a) Potential temperature (at atmospheric pressure), (b) phosphate, and (c) silicate illustrated against salinity. The nitrate concentrations for (d) the summer 2017, (e) fall 2017, and (f) winter 2018 are also shown. The symbols in panels b-f indicate the sample's depth. The open diamonds are near the surface (depth<20-m), while the closed triangles and squares are samples at 25 m and 50 m, respectively. The closed circles represent deeper samples at or below 100 m. Density contours in $\sigma_T$ [kg m$^{-3}$] (a) were computed at atmospheric pressure. Concentrations measured from the most upstream station U37 were 26.43 mmol m$^{-3}$ for nitrate, 0.53 mmol m$^{-3}$ for phosphate, and 39.8 mmol m$^{-3}$ for silicate. This bottle at 10-m depth corresponded to salinity of ≈0.5 psu. $T_f$ is the freezing temperature of seawater in (a).



## 4 Results

### 4.1 Winter conditions

The salinity transect highlight that we visited the HLC during an upwelling event (Figure 3b). The salinity was relatively
homogeneous across the depth when we sampled at L0 during the flooding tide (Figure 2). The water was nonetheless 2 psu
more saline than near-surface waters (i.e., 15 m) measured at stations both upstream and downstream from the HLC (Figure
3b). The higher salinities at the HLC, station L0, cannot be attributed to inputs from the Saguenay Fjord. This waterway
provides a freshwater source, confirmed by the AZMP's annual helicopter survey a few weeks later (Figure 4b). We attribute
the relatively high salinities near the HLC to tidal upwelling and mixing that characterize this region throughout the year
(Ingram, 1983; Galbraith, 2006; Cyr et al., 2015). At the HLC, nitrate concentrations in the upper 50-m were also lower than
water both upstream and downstream at comparable depths (Figure 3a). This presence of low nitrate and high salinity water,
especially in the upper 50-m, further supports that water was being tidally-upwelled.

The transects illustrate more clearly, than the nitrate-salinity diagrams, the presence of a sub-surface nitrate minimum in
the LSLE (Figure 5b). During winter, nitrate concentrations were relatively high near the surface but decreased to reach a
subsurface minimum at the 50-m deep sample. These subsurface samples were near or slightly more saline than 31.2 psu,
the corresponding break between nutrient-rich fluvial waters and the relatively nutrient-poor water downstream seen in the
nitrate-salinity diagram (Figure 4a). In the LSLE, a nitrate-poor subsurface layer is overlaid by fresher and nitrate-rich surface
waters.

### 4.2 Seasonal nitrate variations

We observed this subsurface nitrate-poor layer during other seasons (Figure 5b-d). Climatological averages show that a sub-
surface nitrate minimum is typical of the LSLE's lower reaches during the ice-free months (see Figure 3 of Cyr et al., 2015).
Above this layer, near-surface nitrate concentrations are higher, especially during winter (e.g., $\sim 350$ km in Figure 5b). These
high nitrate concentrations were associated with low salinities (e.g., station L34 and L176 in Figure 4), reflecting the input
of nutrient-rich fluvial water from the USLE. During winter, these fluvial waters likely extended downstream to 400-km from
Quebec City (station L176; Figure 5b), but extended less far in the fall (300 km, Figure 5c). The subsurface nitrate minimum
that the input of fluvial waters creates was more evident in the winter and fall than in summer given the higher consumption
(200-300 km, Figure 5d). However, the nitrate-salinity diagrams show that the fluvial waters may be perceptible just as far
downstream into the LSLE during summer than in fall (Figure 4d-e). Upstream in the USLE, nitrate concentrations were
highest in winter, followed by the fall and summer (Figure 5b-d). Overall, the USLE supplied nitrate-rich fluvial waters into
the surface layer of the LSLE.

The nutrient inventory created in the upper 75-m of the LSLE during the fall and winter measurements is illustrated in Figure
5a. The inventory is obtained by depth-integrating the difference between the fall and winter measurements (Figure 5b and c).
This analysis assumes no nitrate consumption between these two seasons, i.e., no sink term. For completeness, we also added
the inventory created between winter and summer, which is about 25% larger than the nitrate accumulated between winter and





fall. Hence, most of the nitrate accumulates in the surface layer between fall and winter because of the lower consumption rates during these two seasons compared to summer (Figure 16 of Villeneuve, 2020). The LSLE's inventory includes mainly contributions from the nitrate-rich and fresher water originating from the USLE (i.e., $S < 31.2$ psu in Figure 4). The high values at 420 km reflect the nitrate-rich and fresher water from USLE during winter since the near-surface nitrate is relatively depleted during fall at this location (Figure 5a-b). The spatially-averaged inventory between these two seasons was 280 mmol

$m^{-2}$, which translates to an equivalent vertical flux of 32 nmol $m^{-2}$ $s^{-1}$ given the 100 days between the fall and winter measurements. Hence, the minimum nitrate load required to fill the upper 75-m of the LSLE (assuming no consumption or export) was about 195 mol $s^{-1}$.



**Figure 5.** (a) Nutrient inventory generated in the upper 75-m of the LSLE's water column. (b) Nitrate concentrations during our winter field campaign, (c) fall 2017 and (d) summer 2017 monitoring surveys. The thick cyan lines in (b) denote the same water masses as in Figure 3. Water in between these two lines coincides with the subsurface nitrate minimum at station L0 and downstream. The grey circles are scaled with the salinity of the water samples as shown in Figure 4.



### 4.3 Nitrate transport pathways

We now present the annual cycle of fluvial nitrate loads, so they can be compared with the vertical loads entering the LSLE
from vertical mixing processes.

#### 4.3.1 Fluvial nitrate loads

The fluvial nitrate loads vary seasonally, reflecting the changes in dissolved nitrate concentrations and flow rates at Quebec
City. The 17-year long statistical records show that nitrate concentrations reach their annual minimum of $\sim$14 mmol m$^{-3}$ at
the end of summer (Aug and Sept, Figure 6a). Nitrate concentrations progressively increase during fall and remain relatively
steady between December and March. Nitrate consumption is weakest during the winter months (Villeneuve, 2020). In spring,
nitrate concentrations increases, which peak at $\sim$32 mmol m$^{-3}$ in May before steadily decreasing throughout the summer
period. Flow rates follow a similar pattern, albeit the snowmelt generates a more pronounced increase during spring. Flows
peak in April with the average exceeding $16\times10^3$ m$^3$ s$^{-1}$, whereas they reach minimum annual values in late summer with
average flow rates below $10\times10^3$ m$^3$ s$^{-1}$ (Figure 6a). During our 2018 winter campaign, the flow rates were relatively large
compared to the long-term historical averages for February. However, our measured nitrate concentrations were similar to the
February average.

Our observed flows and nitrate concentrations translate into a fluvial nitrate input of $350 \pm 65$ mol s$^{-1}$ at Quebec City.
This value is slightly higher than the 17-year average for February and also higher than the historical average of $300 \pm 20$
mol s $^{-1}$ for the winter months of November through to February (Figure 6b). The fluvial nitrate loads peaked in April when
the historical averages reach 500 mol s$^{-1}$ before decreasing to their annual minimum in August and September. During these
late summer months, nitrate loads are below 150 mol s$^{-1}$ (Figure 6b). These low nitrate loads coincide with the period when
biological consumption is greatest upstream of Quebec City (Hudon et al., 2017) and flow rates are at their lowest. The fluvial
inputs during the field experiment of Cyr et al. (2015) were about 90 mol s$^{-1}$, much lower than historical averages (Figure 6b).
Their fluvial nitrate loads were more than three times less than those during our winter campaign.





**Figure 6.** a) Monthly statistics of the freshwater flow rates at Quebec city from 1995 to 2011 using the inverse modelling techniques described by Bourgault and Matte (2020a). The secondary green axis is the monthly statistics of nitrate concentrations measured at Quebec city during this same period (Hudon et al., 2017). (b) Monthly statistics of the mass nitrate fluvial loads estimated from (1). The bars extend in both panels extend from the 5th and 95th percentile for each month's data and are centred around the median (dark circles). The open circles denote the monthly averages. The error bars on the instantaneous nitrate loads for our winter 2018 campaign and Cyr et al. (2015) late summer observations are the errors propagated from the flow estimates. The nitrate concentrations for Cyr et al. (2015) were interpolated from the temporal time series of Hudon et al. (2017) at Quebec City.



### 4.3.2 Turbulence observations and vertical nitrate fluxes

The vertical nitrate fluxes were determined from the direct turbulence measurements along the Lower St. Lawrence Estuary and Gulf (Figure 7). The turbulence observations were most energetic in the upwelling-driven polynya near the HLC. At station L0, at the sill near the HLC, dissipation $\epsilon$ exceeded $10^{-7}$ W kg$^{-1}$ near the bottom and the surface (Figure 7a), while the mixing rates $K$ exceeded $10^{-4}$ m$^2$ s$^{-1}$ throughout most of the water column (Figure 7b). The other stations were relatively quiescent with $\epsilon$ of the order $10^{-9}$ W kg$^{-1}$. Elevated diapycnal mixing rates were also found in the surface mixed layer (depth <50 m), especially in the Gulf, presumably due to winter convective mixing processes reaching the deeper pycnocline (Figure 7d). Elsewhere in the water column, outside of the energetic HLC, mixing rates were more typical of the ocean interior ($\mathcal{O}(10^{-5})$ m$^2$s$^{-1}$; Waterhouse et al., 2014). Near the HLC, our winter "hot-spot", diapycnal mixing estimates were within the ranges observed during the summer months (e.g., Figure 11 of Cyr et al., 2015). They would have likely been higher if collected at high tide given the late summer observations of Cyr et al. (2015). They measured $K \approx 10^{-2}$ m$^2$ s$^{-1}$ at high tide between depths of 25 and 50 m, compared to $K \approx 10^{-4}$–$10^{-3}$ m$^2$ s$^{-1}$ about 2 hours earlier. Elevated mixing occurred at the HLC and at the base of the surface layer elsewhere in the system.

We obtained the largest vertical fluxes of nitrate near the HLC, where mixing was highest. Nitrate fluxes $F_\mathcal{N}$ were particularly elevated in the deeper part of the water column where they exceeded 30 nmol m$^{-2}$ s$^{-1}$ below 80 m (Figure 7c). This flux converts into a vertical supply of nitrate of about 6 mol s$^{-1}$ into the LSLE's surface layer with the exaggerated surface area of 200 km$^2$ for the HLC (§3.2.3). Further downstream in the LSLE, nitrate fluxes were lower than at the HLC. Vertical nitrate fluxes were generally higher at the interface, separating the sub-surface nitrate minimum. They dropped in magnitude further downstream in the LSLE (station L66 vs L176 at 60-m depth, Figure 7c). Vertical fluxes of nitrate were much weaker in the Gulf than in the LSLE, with weak upward fluxes at the surface layer's base. Occasional storms would increase surface mixing, but its effects would unlikely be felt below 75-m in the LSLE given the stratification imparted by the fluvial waters. Storms, instead, would tend to mix down the nutrient-rich surface water of fluvial origin into the relatively nutrient-poor water beneath it.

In the LSLE, nutrients were transported by turbulence mixing processes both upwards and downwards into the subsurface minima. On average, the nitrate-rich surface waters from the USLE were mixed downwards at 4.8 nmol m$^{-2}$ s$^{-1}$ into this nitrate-poor subsurface layer in the LSLE (50 m depth, Figure 7f). The nutrient-rich water, originating from the Gulf, was mixed upwards into this subsurface layer at a lower rate of 2.4 nmol m$^{-2}$ s$^{-1}$ (80 m depth, Figure 7f). Only the vertical nitrate fluxes from below the sub-surface layer contribute to the LSLE's surface layer's overall supply. When converted into a vertical nitrate load, the relatively weak vertical fluxes outside of HLC supplied 14 mol s$^{-1}$ into the LSLE's surface layer. If we add the contributions at the HLC, the total vertical nitrate load into this layer was of the order of 20 mol s$^{-1}$ during our winter campaign.

We now explore the possibility that our vertical winter loads of nitrate may under-estimate the actual nitrate contributions given the tidal-variability of the mixing at the HLC. For this analysis, we rely on the tidally-resolved summer measurements of Cyr et al. (2015), reproduced in Figure 8 with our winter vertical fluxes. Our winter flux estimate at the HLC was consistent





with theirs collected at the same tidal phase, i.e., 2 hours before high tide (Figure 8a). However, their summer mixing and

vertical nitrate fluxes increased by almost three orders of magnitude during the semi-diurnal tidal cycle with fluxes peaking at

high tide. We converted their vertical fluxes into nitrate loads supplied across the HLC's area in Figure 8b. Their averaged value

of $\sim 1$ $\mu$mol m$^{-2}$ s$^{-1}$ results in a vertical nitrate load of about 210 mol s$^{-1}$. This estimate likely over-exaggerates the vertical

contributions from tidal-upwelling and mixing processes since removing the highest estimate reduces the average nitrate flux

by 50% yielding an equivalent vertical load of 100 mol s$^{-1}$. Their median estimate is even lower than this average, reducing

the vertical nitrate load to 25 mol s$^{-1}$ for the HLC (Figure 8b). All these vertical nitrate loads are still lower than the fluvial

loads of 350 mol s$^{-1}$ entering the LSLE during winter.





**Figure 7.** (a) $\epsilon$, (b) $K$, and (c) $F_N$ after time-averaging the repeated VMP profiles at each station. We present the average profiles for the LSLE (outside of the HLC) and in the Gulf for (d) $\epsilon$, (e) $K$ and (f) $F_N$. The error bands represent 95% confidence intervals obtained through bootstrap analysis on the averages within each region. The HLC's vertical fluxes in (f) are shown on a separate axis. The gray contour lines in (a) and (b) represent isopycnals (in $\sigma_T$, kg m$^{-3}$), whereas those in (c) are the nutrient contours (in mmol m$^{-3}$) shown in Figure 5b. Positive values in (c) indicate an upward flux of nutrients. The thick cyan lines in (c) are the same as in Figure 3 – the 31.2 and 31.9 psu isohalines that delineate water originating from the USLE ($S < 31.2$ psu) and from the Gulf ($S > 31.9$ psu).

**Figure 8.** (a) Nitrate fluxes at the HLC as a function of the tidal phase for winter and summer. The summer estimates from Cyr et al. (2015) are re-illustrated in (b) as a histogram. The error bar shows the 5 and 95th percentile of the observations and is centred around the median $F_N$ (blue is winter and magenta is summer). (c) Close-up of Fig 5b with the estimated nitrate loads from vertical mixing and fluvial contributions. The cyan and green box denotes the extent of the HLC and LSLE, respectively. The secondary y-axis in (b) is the vertical nitrate load after multiplying $F_N$ by the surface area of 200 km$^2$.





## 5 Discussion

We revisit the notion of tidal-upwelling and mixing processes, especially at the HLC, acting as the *nutrient pump* for the LSLE. The low consumption during winter provides a better representation of the physical mechanisms transporting nitrate

into the LSLE than in summer. Summer observations invariably track both physical and biogeochemical processes. Studies have reached variable conclusions about the importance of tidal-upwelling and mixing at the HLC in supplying nutrients to the LSLE, and ultimately the Gulf (see section 1). All of these studies concluded that vertical mixing processes dominate the supply of nutrients in the LSLE (e.g., Steven, 1974; Sinclair et al., 1976; Savenkoff et al., 2001; Cyr et al., 2015; Jutras et al., 2020; Greisman and Ingram, 1977). The biogeochemical box-model of Jutras et al. (2020) predicted vertical nitrate loads of

1050 mol s$^{-1}$ for the entire LSLE, while that of Savenkoff et al. (2001) predicted 685 mol s$^{-1}$. To reach these high values, the turbulent nitrate fluxes at the HLC would need to be two to four times larger than the summer values observed by Cyr et al. (2015), yet their summer fluxes were some of the highest reported in the world (see Table 1 of Cyr et al., 2015).

These biogeochemical studies contrast with our winter observations since the fluvial nitrate loads dominated the supply of nutrients into the LSLE (Figure 8c). Our conclusion remains true even if the winter vertical nitrate fluxes at the HLC reached

values as high as those measured during summer upwelling. During winter, the LSLE's intermediate layer coincides with our observed nitrate subsurface minima. So the estuarine circulation entrains nitrate-rich fluvial waters into the nitrate-poor subsurface layer (Figure 8c). Our direct turbulence observations indicate that the estuarine circulation in the LSLE causes relatively weak entrainment of deep bottom waters into the intermediate layer. The vertical nitrate load caused by this entrainment were about 14 mol s$^{-1}$ — the same magnitude as those obtained by Cyr et al. (2015) during late summer (Figure 8c). The

nitrate-salinity diagram (Figure 4a) and their transects (Figure 5) also highlight that a large portion of the nitrate in the LSLE originates from fluvial sources entering from the USLE.

Our results indicate that the fluvial loads are a much significant input of nutrients into the LSLE than turbulence mixing processes and are transported far into the LSLE. Previous studies have been biased by the dynamics at the end of summer, or may have been too limited in spatial coverage to track the fluvial nitrate-rich waters into the LSLE. For example, Cyr

et al. (2015) focused on the importance of vertical mixing on supplying nutrients into the LSLE's euphotic zone. Their fluxes, although some of the largest fluxes in the world, exceed fluvial loads only during the period they measured – late summer/early fall. Our analysis shows that this period is when the fluvial nitrate loads reach their annual minimum (Figure 6b). In fact, they collected their measurements of vertical fluxes when fluvial loads reached historical lows (Figure 6b). Their average vertical load of nitrate (210 mol s$^{-1}$), for which we exaggerated the area of influence of tidal-mixing processes at the HLC, exceeds

fluvial loads only during summer from August to October inclusively (Figure 6b).

Future field campaigns should focus on documenting the physical processes responsible for upwelling nitrate into the euphotic zone at the HLC. Specifically, estimating the magnitude of the vertical nitrate fluxes during an entire semi-diurnal tidal cycle for both neap and spring tides. Peak tides could pump nitrate into the surface layer at a much higher rate, as shown by Cyr et al. (2015). Differences in magnitude could exist between neap and spring tides (e.g., Sharples et al., 2007; Green et al.,

2019). The goal of these additional measurements would be to remedy the short portion of the tidal cycle resolved during





winter. Our winter observations were nonetheless consistent with the tidally-resolved measurements of Cyr et al. (2015) during summer.

# 6 Conclusions

The inaugural Odyssée campaign aboard the CCGS *Amundsen* icebreaker provided the first winter turbulence measurements in the St. Lawrence Estuary and Gulf. We collected these opportunistic measurements in tandem with the vessel's icebreaking and ship escorting operations. These mixing measurements covered the most considerable spatial extent of the Estuary and Gulf during any season. Our analysis shows that tidal-upwelling appears to be a less effective mechanism for supplying nutrients into the euphotic zone during winter than in summer. In fact, for most of the year, we expect higher nitrate loads from fluvial sources than from tidal-upwelling. In particular, during freshet as the snowpack melts. At this time of the year, both nitrate concentrations and flow rates are highest (Figure 6). The fluvial nutrients loads during winter, a time when biological consumption is low, likely preconditions the phytoplankton spring bloom far into the LSLE and adjacent Gulf. Throughout most of the year, the fluvial contributions from the urbanized catchment upstream of Quebec City are a vital nutrient supply to the St-Lawrence Estuary.

*Data availability.* The presented Odyssée observations are available in the following repository http://doi.org/10.5281/zenodo.3840552. The Canadian government monitoring survey observations can be requested through the St. Lawrence Global Observatory: https://catalogue.ogsl.ca/organization/mpo.

*Code and data availability.* The code and computed flow rates are also publicly available at the codeocean repository (Bourgault and Matte, 2020b).

*Author contributions.* Conceptualisation of winter field campaign by CB, DB and PG. CB collected and processed the winter turbulence data observations, in addition to the formal analysis and data visualisation for the manuscript. VV collected and curated the nutrient observations (Winter 2018 and Fall 2017 from the St. Lawrence Ecosystem Health Research and Observation Network). DB provided the software and hydrological data. PG collected, curated and analysed the AZMP monitoring data. CB prepared the manuscript with contributions from PG and DB, who critically reviewed the manuscript's scientific content. Resources and funding were provided JET, DB, and PG.

*Competing interests.* The authors declare that they have no competing interests.





*Acknowledgements.* This research was funded by the Odyssée Saint-Laurent Program of the Réseau Québec Maritime network, the National

Sciences and Engineering Research Council of Canada (NSERC), and the Canada Foundation for Innovation. The work contributes to the scientific program of Quebec-Ocean. CEB thanks NSERC for funding a fellowship to carry out the work and acknowledges that the research contributes to the project "Quantifying and parameterising ocean mixing" funded by the Australian Research Council (DP180101736). We thank Pascal Guillot from Quebec-Ocean who quality-controlled the CTD aboard the rosette during the Odyssee cruise. We thank the Captain and crew of the CCGS *Amundsen*, and the staff from UQAR, Université Laval, who aided in the collection of data.





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
