# Peer review of "Winter observations alter the seasonal perspectives of the nutrient transport pathways into the Lower St. Lawrence Estuary"

_Ocean Science, 2021_

## Author Comment (AC1)

**REFEREE 1**

**Formal review**

*The manuscript is well written, scientifically sound and addresses relevant societal and scientific questions within the scope of Ocean Science. A nutrient budget analysis based on newly collected and previously published data sets is presented. Unlike previous results, which had suggested that vertical nutrient fluxes due to diapycnal mixing dominate the nutrient supply in the Lower St. Lawrence estuary, the authors conclusively show that fluvial advection of nutrient rich waters from the St. Lawrence River dominate the budget throughout most of the year. The results thus modify current understanding of relevant nutrient supply processes in the estuary.*

*The contribution is well structured and clear. The results fully support the authors interpretations and the description of experiments and calculations are sufficiently complete and precise to allow their reproduction. I also appreciated the open access of scripts and data. Below, I provide a few minor remarks with reference to lines in the manuscript that the authors may want to consider to improve the manuscript. Most of the remarks are related to the discussion of possible biogeochemical processes that could also impact the nitrate budget and the nutrient distributions shown. Throughout the contribution, the authors refer to nutrient "consumption" as the only biogeochemical process being relevant. Although not explicitly defined, it seems that this term refers to uptake of nitrate during primary production. On the one hand, I find the word "consumption" in this context rather unfavorable. In biological oceanography, consumption is widely used related to oxygen and describes the loss of oxygen due to respiration of organic matter, i.e. a chemical reaction. In the same context, nitrate consumption occurs in anoxic waters in the form of denitrification or ANAMOX where bacteria respire nitrogen nutrients instead of oxygen. However, these biogeochemical processes are very different from nutrient uptake during photosynthesis. Thus, I would suggest to replace "consumption" with "uptake" in most places of the manuscript. On the other hand, there is also a biogeochemical nitrate source term. During the degradation of organic matter, nitrification enriches inorganic nitrate concentrations in the water column. While box models suggest that this term in not dominating nitrate supply in the Lower St. Lawrence Estuary, it does seem to contribute between 10% and 20% to the nitrate budget (e.g. Jutras et al., 2020, Thibodeau et al., 2013) and should thus not be completely ignored when interpreting nutrient distributions and their seasonal variability. In my detailed remarks below, I am pointing to a few but not all passages which the authors may want to improve.*

We thank Referee 1 for their feedback, and respond to their detailed remarks below. We have amended the manuscript by replacing the term "consumption" with uptake. As stated in the introduction and methods section, our manuscript focuses on the seasonality of the nutrient transported into the system via physical processes. Specifically, we wanted to draw attention to the large nutrient inputs from fluvial sources given the extensive summer literature that stipulates vertical mixing processes are a main source of nutrients into the surface layer. Besides, we are unaware of any published studies discussing the seasonality of nitrification and remineralization of organic matter. Most studies such as the one referenced by the referee (e.g., *Jutras et al.*, 2020) focus on the summer conditions.

**Detailed remarks with reference to lines in the manuscript:**

*Line 31, "..., but the low nutrient consumption provided a better representation ...". I can understand this statement as far as nutrient uptake during primary productivity is concerned.*

*However, I wonder about the seasonality of biological nutrient sources due to processes such as organic matter remineralization with subsequent nitrification and nitrogen fixation.*

As already stated in the introduction, our manuscript focuses on the physical transport of nutrients into the surface layer. Namely, we compare the fluvial nitrate inputs with those entering the box from vertical mixing processes. We now specifically state that our analysis ignores nutrient cycling in the system or exported nitrate (L60-62). This statement was initially in the methods section.

*Lines 33-34, "upwelling" and "entrainment": I had difficulties understanding the two terms, here. To me, the term "upwelling" refers to vertical advection and involves vertical velocities (e.g. due to Ekman divergence). However, here, I think the authors associate "upwelling" to a vertical flux of nutrients due to diapycnal mixing that does not involve any vertical velocity. Furthermore, the term "entrainment" is unclear to me. How does it differ from mixing? Please clarify the processes that are referred to here.*

We have replaced most instances of "tidal-upwelling" with "tidal-induced mixing" to reduce ambiguities with this term. We used "upwelling" to imply a vertical motion/transport of deep water towards the surface. At the head of the channel, this vertical motion is caused by the tides and the sharp sill, which in turn generates internal waves. We have modified lines 26-34 accordingly. Entrainment is caused by shear (velocity gradients) at interfaces, and mixes water masses (transfers material/water across the interfaces). We have now specified that these processes are associated with vertical mixing processes (L28-29).

*Line 75: add a period after "saltier".*

This typo has been amended.

*Line 175, "The historical dissolved nitrate concentrations at Quebec City were digitized from published sources (Figure 4 of Hudon et al., 2017)." This is a bit unclear. I could not find any nitrate values in Fig. 4 of Hudon et al (2017). There, only the sum of nitrate and nitrite is shown. How were nitrate value derived from this graph? Were nitrite concentrations neglected here? Please clarify.*

Indeed, our observed nitrite concentrations were very low (2-3 orders of magnitude lower than nitrate). We thus relied on nitrate concentrations for our turbulence analysis. We could have used instead the sum nitrate+nitrite, which would have yielded the same results for the fluxes. We have updated the text on L167-170 to stipulate that Figure 4 of Hudon et al. 2017 showed nitrate+nitrite, and that our measured nitrate concentrations are representative of the sum of nitrate+nitrite.

*Line 192, functional dependence of N: The Greek symbol rho is not introduced and should be potential density (otherwise, compressibility needs to be account for in the equation).*

We have modified this line by introducing $\rho$ as potential density.

*Line 213: add potential before density.*

We have removed the mathematical variable, and added the word potential.

> *Line 222, "indicating that nitrate was being consumed": I would suggest to rephrase this sentence to include biological production of inorganic nitrate, e.g. "indicated that nitrate uptake exceeded biogeochemical nitrate sources" or "indicated a net nitrate loss through biogeochemical processes".*

We have rephrased this sentence with "biogeochemical processes caused a net loss of nitrate in the Upper Estuary" [L216-217].

> *Line 229, "Nitrate concentrations in winter ...": I think the statement made in this sentence also applies to nitrate distributions during the other seasons.*

Indeed, this statement may be applicable to other seasons. This statement, however, introduces the topic of the paragraph i.e., to describe the water masses during winter.

> *Line 291, "higher consumption": I think that the authors solely refer to nitrate uptake during phytoplankton growth here. However, there are also other nitrate sinks such as denitrification. What may be the seasonal variability of these processes? Furthermore, I would suggest to use "biological uptake" instead of consumption.*

We have changed the word consumption to "biological uptake", but have not discussed other losses such as denitrification since the manuscript targets the physical inputs of nutrients into the Lower Estuary.

> *Line 306, "Hence, the minimum nitrate load required to ...": This statement is incorrect, as it neglects local nitrate sources e.g. due to aerobic remineralization of organic material (nitrification).*

This statement is correct as it stipulates that local processes and export (e.g., advection out of the region) are ignored, and so the amount stipulated is the minimum load required. To remove ambiguities, we now state that the nitrate in the upper 75-m increased by 195 mol/s instead of stating this layer requires a minimum load of 195 mol/s [L302].

> *Lines 326-327, "... period when biological consumption is greatest": See comments to line 291 above.*

As requested above, this was changed to "biological uptake".

> *Line 373-408, discussion section: I think that adding a few sentences on the relative importance of biogeochemical flux contributions to the nitrate budget would strengthen this work even further.*

We have not incorporated this suggestion other than add some new references on global distribution of nitrate fluxes (e.g. *Mouriño-Carballido et al.*, 2021). An earlier draft included a brief overview of the budget, which we removed because of the lack of winter observations for the other processes (remineralisation, sedimentation, etc). Published biogeochemical models for summer yield vastly different estimates for the nitrate inputs via vertical mixing processes than those we observed. These models often use much higher inputs for fluvial nitrate loads. Thus, we preferred discussing only deviations in the nitrate inputs into the Lower Estuary from physical processes among studies rather than commenting on the other nitrate sinks/sources within the system.

> *Figure 3 caption: Add "winter" between the and field in the first line. In the text before referencing to Figure 3 for the first time you were mentioning historic data and it is thus somewhat unclear which data are shown.*

We have added the word "winter" as requested.

**REFERENCES**

Cyr, F., D. Bourgault, P. S. Galbraith, and M. Gosselin (2015), Turbulent nitrate fluxes in the Lower St. Lawrence Estuary, Canada, *J. Geophys. Res. Oceans*, *120*(3), 2308–2330, doi:10.1002/2014JC010272.

Jutras, M., A. Mucci, B. Sundby, Y. Gratton, and S. Katsev (2020), Nutrient cycling in the lower st. lawrence estuary: Response to environmental perturbations, *Estuarine Coastal Shelf Sci.*, *239*, 106,715, doi:https://doi.org/10.1016/j.ecss.2020.106715.

Mouriño-Carballido, B., et al. (2021), Magnitude of nitrate turbulent diffusion in contrasting marine environments, *Sci. Rep.*, *11*(1), 18,804, doi:10.1038/s41598-021-97731-4.

Savenkoff, C., A. F. Vézina, P. C. Smith, and G. Han (2001), Summer transports of nutrients in the Gulf of St. Lawrence estimated by inverse modeling, *Estuarine, Coastal and Shelf Science*, *52*, 565–587, doi:10.1006/ecss.2001.0774.

Sinclair, M., M. El-Sabh, and J.-R. Brindle (1976), Seaward nutrient transport in the lower St. Lawrence Estuary, *Journal of the Fisheries Research Board of Canada*, *33*, 1271–1277, doi:10.1139/f76-163.

Villeneuve, V. (2020), Caractérisation des variations saisonnière et spatiale des éléments nutritifs et de la prise de l'azote dissous dans lâestuaire du fleuve saint-laurent, MSc thesis, Université Laval.

---

## Author Comment (AC2)

**REFEREE 2**

*The paper by Bluteau et al discusses the nutrient transport from diverse pathways in the Lower St. Lawrence Estuary. From new data collected in winter and previously published data sets, a nutrient inventory in the upper water column is presented. Their results indicate that the fluvial loads are a much significant input of nutrients into the Lower St. Lawrence Estuary than turbulence mixing processes.*

*I think that this manuscript offers very valuable information that revisit the relative importance of fluvial and vertical inputs for supplying nutrients into the St-Lawrence Estuary throughout the year. The rationale of the paper is solid, the methods used are excellent, the analyses are appropriate and the interpretations well-founded. I therefore recommend it for publication*

**Minor comments**

*Please find below some minor points for the attention of the authors:*

*Lines 106-110: subscripts and superscripts should be used in chemical compounds such as NO3-, NO2-, PO4-3 and Si(OH)4*

We have fixed the subscripts and superscripts of these chemical compounds.

*In figure 3, 5, 7 and 8, isolines are shown in the graphs, but what kind of method/interpolation have you used to generate the contour lines?? You should specify it*

The contours were obtained using a Data Interpolating Variational Analysis (DIVA) technique to handle the unevenly spaced spatial data. This algorithm is embedded in Ocean Data View Software. We have now cited the DIVA technique in the captions of each figure, and cited the Ocean Data View software in the code acknowledgements.

---

## Author Comment (AC3)

REFEREE 3

*This manuscript offers a valuable contribution to understanding the physical processes that supply nitrate to the St Lawrence Estuary and provides a new seasonal aspect to the problem using a data set collected during the challenging winter months. The main conclusion is that rivers supply a more important fraction of nitrate to the lower estuary than previously thought - vertical fluxes from deep nutrient rich water only exceed fluvial sources during the summer.*

*Overall the work presented is of good quality and for the most part well written. Ultimately I would like to see it published. However, there are a number of improvements and corrections that I believe should be made before it is accepted, to both elevate the quality of the manuscript and make it as accessible as possible to a wide readership - to the benefit of both the author and journal.*

*My major comments concern (1) better framing of the paper, (2) a more substantial discussion that draws upon a more complete budgeting exercise and (3) the method of calculation of vertical nitrate fluxes.*

*Also, given the conclusions drawn here, I think the title of the paper could be a bit bolder and more informative.*

Thank you for your comments, we have responded to them in our responses below. The scope of our manuscript is the seasonality of the nutrient transported into the system via physical processes. Hence, we have not included a complete nitrate budget. Besides, we lack seasonal data on the nitrate cycling from published sources to complete such a budget. To avoid conjectures, we have opted to include average accumulation rates into the LSLE between summer and fall, and between fall and winter (Fig 8). As for the title, we have modified it to reflect that our perspectives on the main transport pathways of nutrients have changed because of the new winter observations.

**Major comments**

*(1) The introduction to this paper could do a much better job of framing the work that follows and in identifying its unique contribution to the understanding of nitrate supply to the St Laurence Estuary. The majority of the first paragraph for example is around access to the Canadian Coast Guard vessel, information that belongs in the Methods section. I would advise that you consider re-structuring the introduction using the following generic guidance:*

*Spell out what the importance (think global perspective) of the topic you are addressing is. i.e. why should we all care about nutrient budgets in estuaries Lay out (succinctly) what we already know Articulate what we don't know (or can't agree on), why this knowledge gap is important and what has prevented us from tackling it Tell the reader what this manuscript is offering to help address this gap Much of the necessary material for this is there, but not laid out in a way that makes the necessary impact and clearly sets the scene for the work that follows. With some re-writing therefore the introduction could be significantly improved.*

The introduction was modified by removing many sentences about the vessel. However, the paper does not aim to create a nitrate budget for the estuary. It quantifies the amount of nitrate being transported through

physical processes during winter, and revisits the notion that vertical mixing processes trumps contributions from horizontal advection. The introduction was re-organised to highlight the objective of the paper, which was hindered by the lack of winter observations, and lack of spatial coverage in the other season's observations. We also clarified the current debate from the various literature on the system.

> *(2) In my opinion, the opportunity to present and discuss a (back of the envelope) nitrate budget for the lower St Laurence estuary has not been fully exploited. Lines 296-307 discuss a nitrate inventory, yet this is not compared to the total supply rates that are calculated here and summarised in Figure 8c. This is a shame. For example, between fall 2017 and winter 2018 there was a 280 mmol m-2 increase in depth integrated N within the lower estuary, which based on the 100 days between the surveys (and 6000 km2 area) would have required a minimum supply rate of 195 mol s-1 (final paragraph of section 4.2). Based on the riverine (350 mol s-1), tidal nutrient pump (6 mol s-1) and shear within the LSLE (14 mol s-1) that are calculated in this paper (total = 370 mol s-1), and assuming that this is reflective of the period between the fall and winter surveys, approx. 175 mol s-1 of nitrate must have been lost from the LSLE during the same period. What processes (biological and physical) could account for this, e.g. estimated export out of the LSLE given typical velocities in the surface layer?*

An earlier draft included a brief overview of the budget, which we removed because of the lack of winter observations for the other processes (remineralisation, sedimentation, etc). We have decided nonetheless to add the nitrate accumulation rate of 195 mol/s between fall and winter on Figure 8. We have modified the summary figure (8) to highlight the tidal phase at which we measured the vertical fluxes during winter. We had already discussed at great length that the tidal-averaged flux for winter may be as large as those observed in summer. Thus, it's possible that the total winter nitrate inputs are larger than the 370 mol/s estimated by the reviewer. The net loss could thus be larger than 175 mol/s.

We do not have sufficient information (either published, government, tidally-resolved data) for estimating the advection (export) losses from the LSLE. For argument sake, we can presume that the nitrate export in the surface layer is larger during winter than in summer. *Sinclair et al.* (1976) repeated nitrate transects near Rimouski (roughly mid-way in the Lower Estuary) between May and September. They estimated a minimum net export of 7.4 ±1.8 mol/s in August (late summer) and a maximum of 105 ±25 mol/s in May (top 50 m of the water column). It's possible that the net advective export is higher during winter than their 105 ±25 mol/s estimated in May, but unlikely given the weaker net flows near the surface in winter than in spring. We cannot say for sure though, so we have not updated the discussion.

As for biological uptake through photosynthesis, we can extrapolate the uptake experiments of *Villeneuve* (2020) to the entire LSLE. This extrapolation yields a photosynthesis uptake of approx 100 to 120 mol/s during winter, and much higher in summer. We have not commented on the proportion of the winter net loss that is due to physical vs biological processes because of the above stated uncertainties. Crudely, the biological update and horizontal advective loss are of the same order of magnitude ($\sim$ 100 mol/s each).

> *A similar calculation could be performed for the period between summer and fall 2017, albeit relying on summer fluxes reported in Cyr et al. 2015. Given that the total supply in the summer of 313 mol s-1 (river + tidal nutrient pump + shear in Figure 8c) is not hugely dissimilar to the winter (although the balance between sources has shifted) and that (by eye from Figure 5) it looks like the summer to fall increase in depth integrated N might be smaller, biological uptake may be playing a more important role (unsurprisingly). Reconciling that (roughly) with published uptake rates would nicely round off your story and improve the discussion section.*

We have added the accumulation of nitrate between summer and fall into the summary Figure 8 and text [L302-304], which we calculated from Fig 5. This spatially-averaged accumulation rate was approx 95 mol/s in the upper 75-m, so indeed smaller than 195 mol/s between fall and winter. Given the total summer supply of 313 mol/s through physical transport processes (which could be higher during spring tides as mentioned by *Cyr et al.* (2015)), we get an approximate net loss of 218 mol/s. We suspect that the export (loss) via horizontal advection was low given the low surface nitrate concentrations and the low exports (<10 mol/s) estimated by Sinclair et al. (1976) during late summer.

We have very little information about uptake rates during summer. Experiments by *Villeneuve* (2020), if extrapolated to the entire Lower Estuary (upper 75-m, which might be too deep in summer), indicate that the net loss could be attributed entirely to a loss via photosynthesis uptake. Our net loss of 218 mol/s is small compared to published values from biogeochemical box models (e.g., *Jutras et al.*, 2020; *Savenkoff et al.*, 2001). However, published biogeochemical models for summer yield vastly different estimates for the nitrate inputs via vertical mixing processes than those we observed. Some even use very large fluvial inputs (4x larger than the late summer estimates) to force their model. Hence, we prefer discussing only deviations in the nitrate inputs from physical processes among studies, rather than comment on the other nitrate sinks/sources.

*(a few more comments on lines 295-307 can be found below)*

*(3) Given the care that is taken to establish nitrate-salinity relationships and therefore accurate nitrate gradients, I am left a little unsatisfied that the vertical fluxes of nitrate have seemingly been taken at a fixed depth (e.g. 2.4 nmol m-2 s-1at 80 m depth – line 356), as opposed to calculated on an isopycnal representative of the base of the surface layer. In a system that experiences considerable isopycnal heave this would be a much more robust approach.*

The fluxes in the Lower Estuary (outside of the HLC) were taken at the base of the subsurface nitrate minima which coincides with strong gradients of nitrate (Fig 7). The elevated fluxes occur at this depth (Fig 7f), hence our choice. This value sits at the 26.5 kg/m3 isopycnal, which is located at 80±5 m for stations downstream of L0 (outside of the head). These elevated fluxes are representative of the dynamics in the system. We have updated L354-356 to indicate the isopycnal that corresponds to the 80-m depth.

**Specific and more minor comments**

*Line 31 (and others throughout the whole manuscript). Use of the term 'upwelling' doesn't seem quite right here and is a bit mis-leading. My understanding is that oscillatory barotropic tidal currents heave isopycnals and nitrate isopleths up and down the face of the sill at the head of the channel. Turbulence generated over the sill (via e.g. internal wave generation/disspation, hydraulic jumps, Kelvin-Helmholtz instabilities, bottom friction...) then enables nitrate from the deep pool to be (irreversibly) mixed into the surface water. Cyr et al. (2015) uses the term tidal pump, which would be a good replacement for 'upwelling'.*

We used the term tidal-upwelling to describe the vertical transport of material associated with the tides. The dome observable from the salt and nitrate transects (Fig. 3 and 5) highlights that deep water was being upwelled. To avoid confusion with wind-driven upwelling, we have replaced most instances of "tidal-upwelling" with "tidal-induced mixing".

*Line 34. 'Entrainment of deep nutrient-rich water from estuarine circulation...' This is associated with shear-induced mixing that results from the estuarine circulation (and I presume also related to the propagation of internal waves?). In the re-working of the introduction, it would be good to provide a clearer mechanistic description of these processes.*

Yes, this sentence refers to entrainment from shear-induced mixing at the interfaces of water layers even though internal waves may be present elsewhere in the Lower Estuary (e.g., wind can generate internal waves). We have included a reference to internal waves in the previous sentence, and removed most instances of "estuarine circulation" in the introduction. The intent here was to list the different physical processes, which are discussed further in Section §2.1 (site description).

*Line 43. Vertical nitrate fluxes in the Mauritanian upwelling region are the highest ever reported then? If so, it would be good to make this more explicit in this sentence.*

This line was updated to highlight that the summer estimates were compared to the tabulated vertical nitrate fluxes of *Cyr et al.* (2015) from numerous studies around the world. The Mauritanian is well known for its productivity [L36-38], and its fluxes are very large but not the largest in the world which is why we did not state this in the text.

*Line 45. Please make clear what process the 33 and 400 mol s-1 refers to.*

This sentence re-states *Cyr et al.* (2015) summer estimates of nutrient fluxes. We have reworded since this sentence since it represents their observed range nitrate transported from vertical mixing processes in the LSLE [L40].

*Figure 1. Although the stations are marked with 'U..' and 'L..', The Upper- and Lower-St Laurence Estuary could be a little more clearly marked.*

We have now labelled the Upper and Lower Estuary in Figure 1b.

*I understand the need for acronyms, but from a readability point of view, especially for those not so familiar with the area, you may like to consider cutting down on the use of HLC, LSLE and USLE, particularly early on in the manuscript. I do not suggest removing the acronyms all together, but a more blended approach might be helpful. Replacing LSLE and USLE with 'Lower- ' and 'Upper-estuary' would work fine in a lot of instances (also HLC with head of the channel) and make the manuscript much easier to read.*

Agreed, we have replaced all in-text instances of LSLE and USLE to Lower and Upper Estuary, respectively. We have also reduced the use of HLC for denoting the head of the Laurentian channel by referring it to the "head" of the "head of the channel".

*Line 75. 'The magnitude and impact of this nutrient transport process on primary production across the whole LSLE and Gulf system is debated'. Can you be a bit more explicit about what is in debate here – material more for the introduction. (ps. this is an instance where you might like to consider writing Lower St Laurence Estuary in full, rather than the acronym).*

This statement refers to the debate of whether the vertical mixing processes dominate the nutrient supply into the Lower Estuary, and more specifically the importance of the intense tidally-induced mixing at the head relative to the vertical mixing elsewhere in the estuary. We have re-written the sentence accordingly and moved it to the introduction, while another sentence was added to highlight that previous observations have focused on mostly on vertical transport processes rather than the fluvial loads [L28-31].

> *Line 166. Is this the right way around? Do you instead mean that the nitrate supply at Quebec City is representative of what reaches the Lower estuary?*

Indeed, we meant the downstream end of the Upper Estuary or the upstream end of the Lower Estuary, but not the upstream end of the Upper Estuary. We have corrected this typo.

> *Liner 185. More specifically the 'mixing rate K' is the vertical eddy diffusivity.*

We used the term mixing rate to denote the diapycnal mixing rates K, which is the same word used in the subsection heading.

> *Figure 4. Might be helpful to include vertical lines (at least in panel f) that mark 31.2 and 31.9 psu.*

We have added dotted vertical lines in panel (f) of Figure 4.

> *Line 264-265. 'Exaggerate'? This isn't especially well articulated. Please can you re-word. You mean upscale the vertical nitrate fluxes observed within the HLC based on an area of 200 km2.*

We re-stated that this 200 $km^2$ is larger than the 100 $km^2$ value used by the summer study of *Cyr et al.* (2015). Hence, using the largest area exaggerates the impact of the vertical fluxes.

> *Line 273. Do you really mean Figure 4b?*

No. Thanks for pointing out this typo created when we re-organised these panels. We have now rechecked all references to sub-panels.

> *Line 279. Figure 5. Small detail, but you might want to consider re-ordering the panels in Figure 5 so that they are referred to in order throughout the manuscript. It would be worth checking that this is also the case for other multi-panel figures.*

The sub-panels were placed in their respective locations to make the figures more intuitive, or to optimise the use of existing axes. We have left the panels as is.

> *Line 282. Figure 4a isn't the nitrate-salinity diagram. Do you mean Figure 4f?*

Indeed. This statement refers to panel f, and this typo has now been rectified.

> *Lines 296-307. As alluded to above, I think that this is material for the discussion section and is not yet used to full effect. Further to the suggestions above I'd emphasise the need to provide indicators of the magnitude of potential biological source and sink terms, e.g. what might remineralisation of organic matter over these months contribute?*

We are unaware of winter observations about these biological terms, so prefer avoiding conjectures about their magnitudes. Current studies for summer either prescribe fluvial inputs that are representative of spring (e.g., *Jutras et al.*, 2020), or predict unrealistic contributions from vertical mixing processes. Our paper thus focuses on improving the magnitude of these physical input terms as stated at the end of our introduction.

> *Line 304 – it would be worth re-stating the area over which the averaging was performed. Also, state the seasons you are referring to with the statement 'between these two seasons' – it is rather inferred, but not crystal clear.*

We added the Lower Estuary to identify the region over which the data was spatially-averaged [L300]. We also brought the words "between fall and winter" to the beginning of the sentence rather than having the seasons specified at the end of the sentence.

> *Query use of term 'load' in some contexts, e.g. line 306-307– is 195 mol s-1 not a supply rate?*

The word 'load' refers to a mass input of a substance as opposed to a concentration, and is a common term in engineering and water quality. Here, we converted them into moles as opposed to mass in grams. We thus retain the term loads for inputs of nutrients.

> *Line 322. I presume (based on the following sentence) that the 350 mol s-1 is for February 2018 - but this needs making clearer.*

We added the words February 2018 to this sentence as referenced in the previous sentences.

> *Line 334. Looking at Figure 7e (yellow profile) I'd say that K exceeded 10-4 m2 s-1 throughout the lower half of the water column, but not across 'most' of it. The upper 60 m is notably less than 10-4 m2 s-1.*

This sentence was re-phrased to provide the depth range over which this statement is true i.e., below 60-m and above 30-m depth of the water column.

> *Line 339-340. Can you be clearer as to which estimates of K might have been higher if collected at high tide – those in Cyr et al. 2015 or your own? From Fig 2 I thought that VMP profiles at L0 were collected at high water?*

The sentence now states on L335 "our winter fluxes", which we collected 2 h before high tide at Rimouski (not at high tide). This information was shown in Fig 2, albeit the entire two-week tidal signal was illustrated. We had also plotted the timing of our winter fluxes relative to the tidal phase in Figure 8.

> *Line 344-346. The calculation here is of the vertical flux across a 200 km2 area, assumed to be representative of the HLC - the assumption then is that this N supply is distributed across the whole LSLE – a much wider area downstream. The scentence needs re-wording to better articulate this (if this is indeed what is meant). This is the vertical flux associated with the processes operative at the head of the channel – i.e. tidal pumping.*

The sentence has been simplified on L343-344 since it converts the vertical fluxes of nitrate at the head of the channel to a net transport at the head. We have assumed the fluxes measured at the head are representative of the 200 km$^2$ at the head. This flux is not representative of the fluxes elsewhere in the LSLE.

> *Figure 8 – panel (c) should be placed below (a) and (b)*

We modified the panel locations in Figure 8 as suggested.

> *Please thoroughly check all the references. There are multiple examples of where the Journal name and/or article doi is missing.*

Missing doi and acronyms of journal names were fixed.

> *Line 73. 'milder inflow'. You mean 'weaker'? 'Milder' is a slightly odd word to use here.*

Indeed. The word milder was replaced with weaker.

> *Line 75. Full stop after 'becomes saltier'.*

We have added the missing stop after saltier.

> *Line 96. '...and nutrient concentrations...'*

It's unclear what this comment refers to. These lines discuss the CTDs used aboard the ship.

> *Line 106 (and others). Super and subscripts for NO3- etc*

These sub/superscripts have been fixed.

> *Line 241 – 'concentrations FROM the vmp...'*

It's not clear what the reviewer is alluding to with this comment. The word vmp is not in these sentences, and no concentrations are being measured by this instrument.

> *Line 269. First sentence needs re-wording.*

This sentence was shortened to introduce the paragraph [L266].

> *Line 291. This sentence doesn't end very clearly. Can you re-word slightly to make clearer that you mean that nitrate uptake in surface waters during the summer is higher than during winter and fall.*

We have amended this sentence to focus on the higher uptake during summer rather than the reduced uptake during fall and winter [L286-288].

**REFERENCES**

Cyr, F., D. Bourgault, P. S. Galbraith, and M. Gosselin (2015), Turbulent nitrate fluxes in the Lower St. Lawrence Estuary, Canada, *J. Geophys. Res. Oceans*, *120*(3), 2308–2330, doi:10.1002/2014JC010272.

Jutras, M., A. Mucci, B. Sundby, Y. Gratton, and S. Katsev (2020), Nutrient cycling in the lower st. lawrence estuary: Response to environmental perturbations, *Estuarine Coastal Shelf Sci.*, *239*, 106,715, doi:https://doi.org/10.1016/j.ecss.2020.106715.

Mouriño-Carballido, B., et al. (2021), Magnitude of nitrate turbulent diffusion in contrasting marine environments, *Sci. Rep.*, *11*(1), 18,804, doi:10.1038/s41598-021-97731-4.

Savenkoff, C., A. F. Vézina, P. C. Smith, and G. Han (2001), Summer transports of nutrients in the Gulf of St. Lawrence estimated by inverse modeling, *Estuarine, Coastal and Shelf Science*, *52*, 565–587, doi:10.1006/ecss.2001.0774.

Sinclair, M., M. El-Sabh, and J.-R. Brindle (1976), Seaward nutrient transport in the lower St. Lawrence Estuary, *Journal of the Fisheries Research Board of Canada*, *33*, 1271–1277, doi:10.1139/f76-163.

Villeneuve, V. (2020), Caractérisation des variations saisonnière et spatiale des éléments nutritifs et de la prise de l'azote dissous dans lâestuaire du fleuve saint-laurent, MSc thesis, Université Laval.

---

## Author Response (AR2)

**Responses to Editor Mario Hoppema**

Dear Dr. Bluteau and co-authors,

Thank you for the revised version of your manuscript. I am satisfied with your response and revisions and your manuscript is now accepted for publication in Ocean Science. There are some final technical issues which are listed below. Please account for these when submitting the final version of your manuscript.

**Thank you Editor Mario Hoppema for your comments. We have corrected the manuscripts as requested. Our respondes are bolded**
* * *
L59 delete Below - **Corrected**

L91 8 to 23 February 2018 (date format) - **Corrected**

L104-105 psu is not a unit and should not be used. Please change wording, using practical salinity or similar
**Corrected this sentence by stating practical salinity at the beginning of the sentence.**
* * *
L108 The notation of nitrate and nitrite as shown here is not clear. Do you mean NO3+NO2? **Yes, we have fixed this passage of text.**

L110 What is +NO2- ? -- **Calorimetric methods measure N03 + N02, and N02. We have fixed the text so this becomes clearer**
* * *
L182 Please do not use psu because salinity does not have a unit (also at many other places in the text)

Figure 3 caption: Please delete psu
L237, 238, 241 no psu
**We searched and replaced psu instances in the text.**
* * *
L474 volume missing -- **Corrected**

L487 double doi
L509 double doi
**We manually removed the url/eprint fields in the bibtex entries. The copernicus bst file should be changed to avoid displaying these additional fields that are downloaded automatically from publishers.**

L549-550 Several strange symbols. Please give correct reference **The accents were fixed in the bibtex entry.**